



# Vertical wake deflection for floating wind turbines by differential ballast control

Emmanouil M. Nanos[1], Carlo L. Bottasso[1], Dimitris I. Manolas[2], and Vasilis A. Riziotis[2]

[1]Wind Energy Institute, Technische Universität München, 85748 Garching b. München, Germany
[2]School of Mechanical Engineering, National Technical University of Athens, 15780 Athens, Greece

**Correspondence:** C.L. Bottasso (carlo.bottasso@tum.de)

**Abstract.**

This paper presents a feasibility analysis of vertical wake steering for floating turbines by differential ballast control. This new concept is based on the idea of pitching the floater with respect to the water surface, thereby achieving a desired tilt of the turbine rotor disk. The pitch attitude is controlled by moving water ballast among the columns of the floater.

This study considers the application of differential ballast control to a conceptual 10 MW wind turbine installed on two platforms, differing in size, weight and geometry. The analysis considers: a) the aerodynamic effects caused by rotor tilt on the power capture of the wake-steering turbine and at various downstream distances in its wake; b) the effects of tilting on fatigue and ultimate loads, limitedly to one of the two turbine-platform layouts; and c) for both configurations, the necessary amount of water movement, the time to achieve a desired attitude and the associated energy expenditure.

Results indicate that —in accordance with previous research— steering the wake towards the sea surface leads to larger power gains than steering it towards the sky. Limitedly to the structural analysis conducted on one of the turbine-platform configurations, it appears that these gains can be obtained with only minor effects on loads, assuming a cautious application of vertical steering only in benign ambient conditions. Additionally, it is found that rotor tilt can be achieved in the order of minutes for the lighter of the two configurations, with reasonable water ballast movements.

Although the analysis is preliminary and limited to the specific cases considered here, results seem to suggest that the concept is not unrealistic, and should be further investigated as a possible means to achieve variable tilt control for vertical wake steering in floating turbines.

## 1   Introduction

Power production from wind is typically organized in clusters of turbines, forming a wind plant. By interacting through their
wakes within the plant, turbines are subjected to adverse effects that reduce their power capture and life expectancy, both for onshore and offshore installations. While in the latter case typical spacings between turbines are quite large, wake-induced losses can still be significant. In fact, in typical offshore conditions wakes persist many diameters downstream of the rotor because of the low turbulence of the atmospheric boundary layer (Vermeer et al., 2003).





Several remedies against these effects have been proposed so far, as for example changing the induction factor (Steinbuch
et al., 1988), redirecting (or "steering") the wake path in the lateral or vertical directions (Parkin et al., 2001; Fleming et al.,
2015; Campagnolo et al., 2016a; Fleming et al., 2019; Campagnolo et al., 2020; Doekemeijer et al., 2021), dynamically exciting
the wake to enhance mixing (Frederik et al., 2020b, a), and various possible static and/or dynamic —largely unexplored—
combinations thereof (Cossu, 2020c). Among these techniques, it appears that static induction is not very effective as far as
power capture is concerned (van der Hoek et al., 2019). On the other hand, dynamic mixing techniques are promising, but
further research is needed to address various concerns related to increased loading and actuator duty cycle (Wang et al., 2020)
and to loss of effectiveness in turbulent inflows (Munters and Meyers, 2018). Among these various proposed solutions, static
wake redirection is the most mature wind farm control technique available today, which has already been demonstrated in field
experiments (Fleming et al., 2019, 2020; Doekemeijer et al., 2021) and it is also offered as a market product (Energy, 2019).

Wake redirection is based on purposely misaligning the rotor with respect to the wind vector, thereby creating a force
component normal to the wind direction that is responsible for deflecting the wake. *Lateral* wake deflection is based on yawing
the turbine out of the wind. Since horizontal axis wind turbines are already equipped with active yaw control, this method does
not require any radical hardware modification. This fact, together with the significant wake displacements that can be achieved
without excessively increasing the loads on the steering turbine, is one of the reasons for the success of this technique, which in
fact has been successfully implemented on wind turbines originally designed without taking wake steering into consideration
(Fleming et al., 2019, 2020; Doekemeijer et al., 2021).

*Vertical* wake deflection works in the same way as lateral deflection: when the rotor is tilted about an horizontal axis, its
thrust is inclined with respect to the ground; in turn, the equal and opposite reaction on the flow is also inclined, resulting in a
vertical force component with respect to the ground that deflects the wake in the vertical direction.

There are, however, some key differences between the lateral and vertical deflection strategies.

First, contrary to lateral wake steering, standard wind turbine configurations do not offer an already existing mechanism that
can be employed for deflecting the wake vertically. The only exception is the case of downwind teetering rotors, where vertical
wake deflection can in principle be achieved by tilting the tip-path plane through blade flapping; however, there are no large
downwind teetering rotors on the market today.

Second, vertical steering presents a strong directional dependence. While also lateral steering is not exactly symmetric
between left and right misalignments because of the rotation of the wake (Fleming et al., 2018), deflecting the wake towards
the sky or towards the ground has profoundly different effects. In fact, in vertically sheared flows, an upward vertical deflection
moves the wake into a higher speed flow region, whereas the opposite happens for a downward deflection. Furthermore, when
subjected to a downward deflection the wake eventually interacts with the ground, resulting in a strong deformation of the wake
structures and in its accelerated recovery (Scott et al., 2020).

Notwithstanding the technical difficulty of implementing vertical wake deflection, this concept has received some attention
in the recent literature. For example, Srinivas et al. (2012) presented an analytical study of vertical steering, and evaluated some
engineering models in their ability to predict the vertical motion of the wake. Fleming et al. (2015) used computational fluid
dynamics (CFD) to simulate a single column of two wind turbines with tilted rotors, while in the paper of Annoni et al. (2017)





the authors simulated up to three turbines in a column; both studies reported significant power gains at the cluster level, caused
by an improved capture downstream that offsets more limited losses upstream. Simulation studies on more complex layouts
were conducted by Cossu (2020a) and Cossu (2020b), where the front two rows of turbines in a farm were tilted, obtaining
significant power gains at the wind plant level. The author also studied the effect that rotor size, longitudinal spacing between
the turbines and thrust setting can have on the plant power output. These studies have highlighted an interesting phenomenon,
whereby downward wake deflection leads to the creation of high-speed streaks in the flow, which again are associated with
significant power boosting at the plant level. Su and Bliss (2019) used a free-wake method to study a tilted rotor, reporting
power benefits for a two-turbine column when deflecting the wake of the upstream machine towards the sky. Scott et al. (2020)
performed wind tunnel measurements of a four-by-three grid layout using scaled wind turbine models, where the machines in
the third row were tilted. Among other results, the authors reported a faster wake recovery for downward deflection than in the
upward case.

In summary, the literature already reports a significant body of evidence indicating that vertical wake steering can be an
effective form of wind farm control. Further potential gains could be possibly achieved by combining vertical and lateral
steering, although this problem does not seem to have been explored yet. However, the problem of how vertical steering can be
achieved in practice remains at present unsolved, except for the downward teetering turbine configuration.

To address this gap, Nanos et al. (2020) proposed a novel way of implementing vertical wake steering for floating wind
turbines. This new idea exploits the fact that semi-submersible platforms, which are among the most popular floating concepts
(Liu et al., 2016), require the use of ballast to achieve a desired attitude with respect to the sea surface. Attitude control is
commonly obtained by storing and distributing water among the columns of the platform in order to change the center of
gravity position. Active ballast control systems are already installed on board semi-submersible platforms used by the oil
industry; the same concept is included in some offshore-wind conceptual designs (Roddier et al., 1997), where its purpose is
mainly to counteract the pitching moment created by the thrust force of the rotor. With reference to wind farm control, the idea
pursued here is to use an active ballast control system to pitch the platform, this way achieving a desired tilting of the rotor disk
and, therefore, a vertical deflection of the wake. The concept of vertical wake deflection through platform pitching by active
ballast control is illustrated in Fig. 1.

The scope of the present work is to refine the concept first presented in Nanos et al. (2020). The objective here is clearly not
to design an actual system implementing vertical wake steering by active ballast control, but to perform a feasibility analysis,
with the goal of answering the following basic questions:

– Is it conceptually feasible to use differential ballast control to perform vertical wake steering for wind farm control? and
 what would be the most and least favourable configurations and operational conditions?

– Can an existing ballast control system be used for this additional purpose (similarly to what has been done with yaw
 control for lateral wake deflection), or should the system be modified somehow?

– Would such a system be able to reach sufficiently large pitch motions (and, hence, rotor tilt angles)? and what would be
 the achievable tilt rate and associated energy cost?



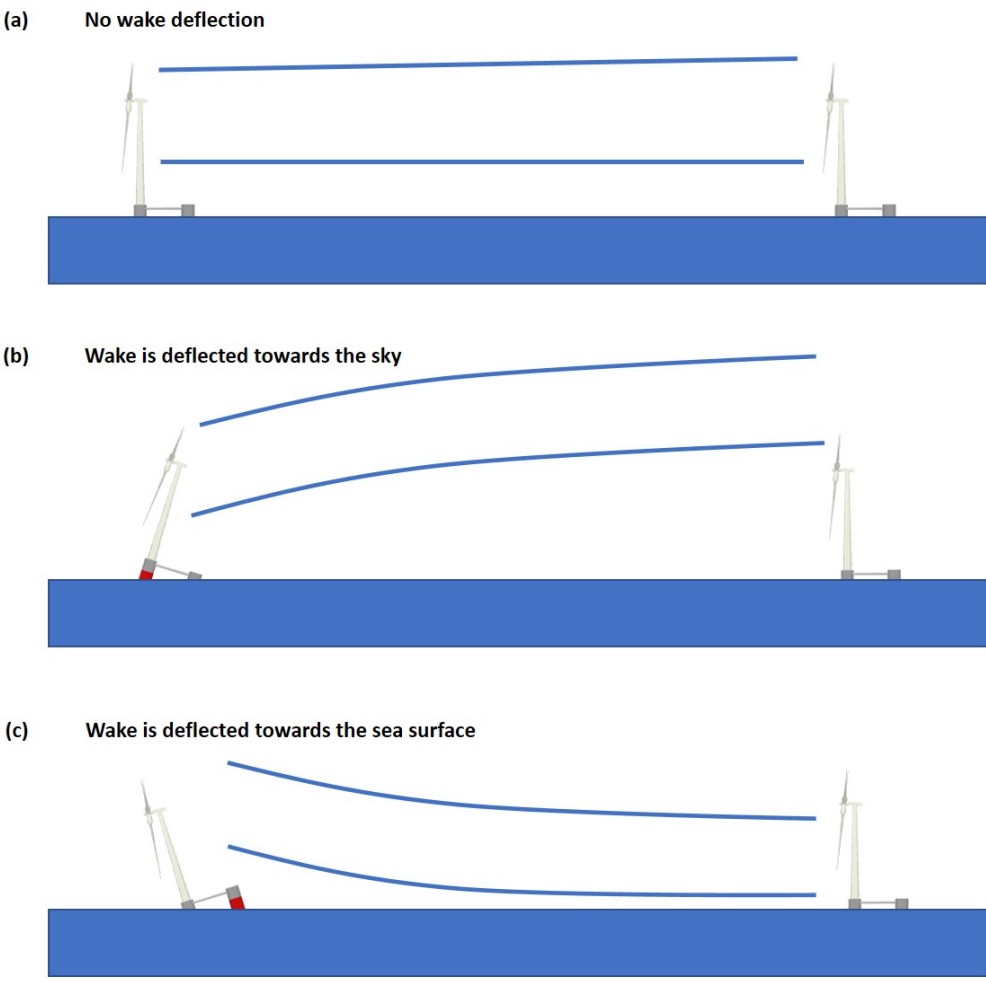

**Figure 1.** Schematic representation of the vertical wake redirection concept.

- Could an existing floating wind system be used for vertical wake steering by ballast control, or would the turbine, platform and/or mooring system need to be partially resized?

More specifically, this paper will:

- Assess the effect of rotor tilt on the wake of a turbine and on the power yield of a column of two turbines. To this purpose, CFD simulations of a scaled wind turbine, validated with wind tunnel experiments, are employed. Expanding on previous tilt misalignment studies, which analyzed streamwise spacings of 6-8 rotor diameters (Fleming et al., 2015; Annoni et al., 2017; Cossu, 2020b; Scott et al., 2020), the present work considers distances up to 12 rotor diameters, which is a more realistic spacing for offshore-wind applications.





– Make preliminary calculations of the quantity of water ballast that needs to be redistributed for achieving the necessary tilting of the rotor, along with estimating the associated energy expenditure.

– Assess the impact of the proposed method on the structural loading of the turbine, the platform and its mooring system.
Although semi-submersible platforms and turbines are designed and certified to withstand significant pitch excursions under extreme weather conditions, a specific assessment of the effects on the structure of this new form of wind farm control is important to evaluate the overall feasibility of the concept. To this end, hydro-aero-servo-elastic simulations of a conceptual wind turbine on a floating platform are utilized.

The article is organized as follows. Section 2 gives a description of two reference platforms and one wind turbine that are used for assessing the feasibility of the proposed concept. Section 3 analyzes the effect of tilting the rotor on the turbine
wake and on the power of a two-turbine cluster through CFD simulations, which were first validated against experimental data. Section 4 presents the effects of tilt on the extreme and fatigue loads computed by hydro-aero-servo-elastic simulations. Section 5 assesses the differential ballast control concept on the basis of an hydrostatic analysis, and presents an initial rough sizing of the system for the two different platform configurations. Finally, Sect. 6 presents the main conclusions and outlines possible future steps.

## 2 Reference turbine and platforms

The present analysis is based on one reference wind turbine and two reference floating platforms. Unlike other wake control strategies, ballast control for vertical wake deflection is substantially affected by the design characteristics of the turbine and of the platform.

Regarding the turbine configuration, upwind wind turbines are favored when wake deflection towards the sky is considered,
because of the built-in uptilt used to increase rotor-tower clearance (Burton et al., 2001). In fact, since the rotor plane is already tilted nose-cone up (typically by about $5°$), in order to achieve a given misalignment a smaller additional rotation is needed for a nose-up attitude (upward wake deflection, as in Fig. 1b) than for a nose-down one (downward wake deflection, as in Fig. 1c). Therefore, smaller platform rotations are necessary for deflecting the wake towards the sky than towards the sea surface. Exactly the opposite happens for a downwind turbine, where the built-in uptilt used to increase rotor-tower clearance
favours downward wake deflections, resulting in smaller platform angles when the wake is steered towards the sea surface than towards the sky.

As shown by previous research (as for example Cossu (2020a, b); Scott et al. (2020)) and later on in this paper, it appears that downward wake deflection is more effective for improving cluster power than upward deflection. From this point of view, a downwind turbine configuration appears to be better suited for this application than an upwind one. Notwithstanding
this important difference between the two configurations, an upwind turbine is used in this work, since it represents today's standard offshore configuration and no large downwind turbines are at present available on the market. The following analyses are based on the DTU 10 MW turbine (Bak et al., 2013), whose basic characteristics are reported in Table 1.



**Table 1.** Basic characteristics of the DTU 10 MW reference wind turbine.

| Data | Value | Data | Value |
|---|---|---|---|
| Configuration | Upwind | Wind class | IEC 1A |
| Rated electrical power | 10.0 MW | Rated thrust | 1 400 kN |
| Hub height [H] | 119.0 m | Rotor diameter [D] | 178.30 m |
| Rotor uptilt angle | 5.0° | Total weight | 1 280 tons |

**Table 2.** Basic characteristics of the two reference platforms.

| | Platform A | Platform B |
|---|---|---|
| Column length | 38 m | 65 m |
| Column diameter | 12 m | 15 m |
| Column to column distance | 56.4 m | 45 m |
| Total weight | 7000 tons | 30 000 tons |

Since the present wake control concept is based on the pitching of the whole platform, also the characteristics of the floater —in addition to those of the turbine— play an important role. In fact, the ballast distribution that is required for a specific pitch attitude depends on the size, weight and draft of the platform. Additionally, ballast is affected by where the turbine is located with respect to the platform, either close to its center or to its edge. In case the platform is not axially symmetric about the turbine tower, the yaw orientation of the turbine with respect to the platform also plays a role in determining the differential ballast that is necessary for a given attitude. Finally, it should be noted that, depending on the configuration of the system, a change of platform pitch can imply —in addition to a tilting— also a vertical motion of the hub; as a result, the rotor can be exposed to a slightly different wind speed in a sheared inflow.

In order to explore some of these configuration-related effects, the present paper makes use of two different reference platforms. The first one, hereafter called Platform A (Fig. 2a and 2b), is based on the concept developed in the WINDFLOAT project (Roddier et al., 1997). The platform is composed by three columns made out of steel, arranged in a triangular configuration by connecting trusses, with the turbine directly placed on top of one of the columns. The second platform (Fig. 2c and 2d) is the tri-spar floater developed in the INNWIND project (Azcona et al., 2017; Manolas et al., 2018), hereafter called Platform B. This floater was developed to accommodate a 10 MW machine mounted on a steel structure at the center of three columns, and it represents a hybrid configuration between a semi-submersible and a large-draft spar buoy. This design uses concrete for the spars, resulting in a much heavier structure compared to Platform A. The principal characteristics of the two platforms are summarized in Table 2.

Figure 2b defines also the angle conventions adopted in this paper. Tilt indicates the angle between the rotor axis and the wind vector, while pitch refers to the angle between the platform and the water surface. While wind vector and water surface



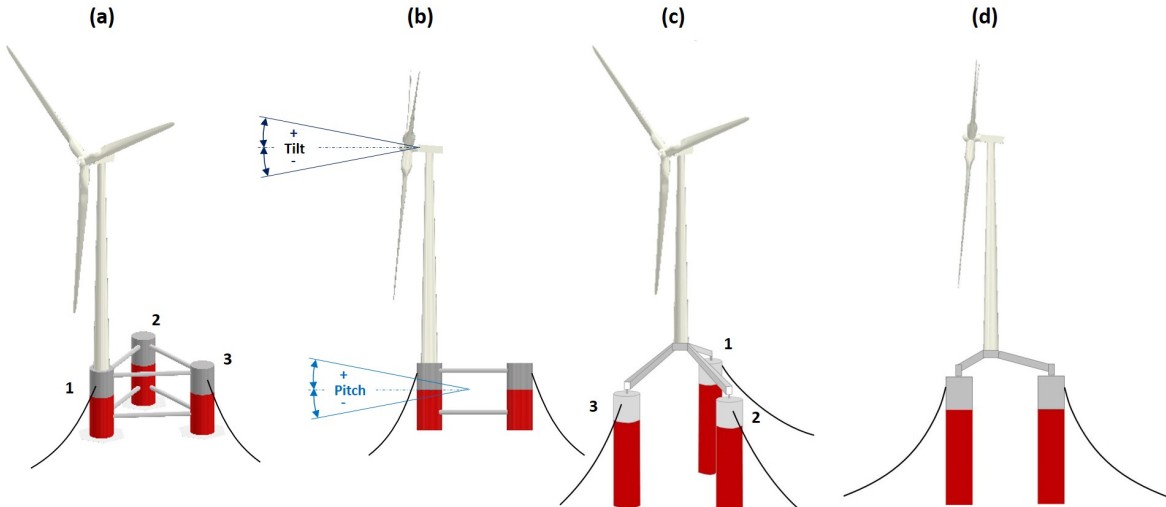

**Figure 2.** Sketch of Platform A, based on the WINDFLOAT concept (Roddier et al., 1997) **(a,b)**. Sketch of Platform B, based on the INNWIND concept (Azcona et al., 2017; Manolas et al., 2018) **(c,d)**.

are assumed to be always parallel, the tilt and pitch angles are not necessarily equal to each other, because of the built-in uptilt used in wind turbines to increase the clearance between rotor and tower with the purpose of relaxing the stiffness requirements on the blade. For a positive tilt, the rotor axis is above the horizon when looking upstream. Hence, an upwind wind turbine

has a positive built-in uptilt, whereas a downwind machine has a negative one; additionally, positive tilt implies that the wake is deflected towards the sky, whereas for negative tilt the wake is steered towards the sea surface. Pitch follows the same sign convention. For better readability, instead of referring to positive and negative angles, the text will refer to *wake-up* and *wake-down* angles, respectively, which is a more intuitive terminology.

## 3    Characterization of the wake

The effects of rotor tilt on wake development and downstream power capture were evaluated by a combined simulation-experimental study. The G06 scaled model (Nanos et al., 2021) of the reference 10 MW wind turbine is used for this purpose. Previous work by Wang et al. (2021) has shown that scaled wind turbines, designed according to the same specifications of the G06 model, are capable of producing highly realistic wakes in atmospheric boundary layer wind tunnels. A CFD simulation model of the turbine was first verified based on experimental measurements performed in the wind tunnel in rotor-tilted

condition, and then used to explore the characteristics of the wake.



### 3.1 CFD validation and set-up

CFD simulations were executed with a flow solver based on a large eddy simulation (LES) actuator line method (ALM) implemented in Foam-extend (Jasak, 2009), while the wind turbine lifting-line aerodynamics was modeled in FAST (Guntur et al., 2016). This framework has been validated in previous work (Wang et al., 2019), and it is further verified here in tilted conditions using new ad hoc wind tunnel measurements.

An experimental campaign was conducted in the BLAST atmospheric boundary layer wind tunnel at the University of Texas at Dallas (UTD). Further details on the wind tunnel and the G06 scaled model are available in Nanos et al. (2018, 2019, 2021). The model was operated at its optimum tip speed ratio and pitch angle. With the help of a tilting mechanism inserted between the nacelle and the tower top, the rotor was tested at three different attitudes: $0°$, $20°$ wake-up, and $-20°$ wake-down.

The wake of the G06 was measured on a vertical plane at a 5D downstream distance by Stereo Particle Image Velocimetry (S-PIV). The velocity at hub height was approximatively equal $10 \ \mathrm{ms}^{-1}$, the turbulence intensity was $8.5\%$, and the inflow had a vertical shear characterized by an exponent $\alpha = 0.2$.

A first set of CFD simulations mimicked the experimental set-up, including the tilting geometry, the wind tunnel walls and the passive generation of the turbulent and sheared inflow, which was obtained by spires placed at the chamber inlet and roughness elements on the floor. Figure 3 shows the simulated and measured vertical profile of the inflow at the turbine location, which appear to be in very good agreement with each other. Figure 4 shows the absolute percent error between CFD and S-PIV measurements for the three tilt angles at a $x/D = 5$ downstream distance. The error was calculated according to the following formula:

$$\epsilon = \left| \frac{u^{\mathrm{exp}}/U^{\mathrm{exp}}_{\mathrm{hub}} - u^{\mathrm{cfd}}/U^{\mathrm{cfd}}_{\mathrm{hub}}}{u^{\mathrm{exp}}/U^{\mathrm{exp}}_{\mathrm{hub}}} \right| 100, \tag{1}$$

where $u$ is streamwise velocity in the wake and $U_{\mathrm{hub}}$ the inflow velocity at hub height. Results show that the error is for the most part of the domain between $0\%$ and $2\%$, reaching $4\%$ in some limited areas. This error is considered acceptable for the scope of the present analysis. Additional details on the experimental set-up, the S-PIV data and the comparison between experiments and simulations are available in Nanos et al. (2020).

### 3.2 Effects of tilt on the flow

After validating the CFD model for this specific set-up, additional simulations were conducted at different rotor tilt angles. This second set of simulations was based on the configuration of Fig. 2a,b, where the rotor is facing away from the other two columns. This implies that, since the position of the turbine for each tilt angle is determined by the platform kinematics, a wake-up tilt rotation generates a small vertical lifting of the hub, whereas a wake-down tilt comes with a small downward motion.

To compute the power available in the wake, the power coefficient of the untilted configuration was obtained from the CFD results by computing a rotor-effective wind speed. Next, using the computed value of the power coefficient, the power in the wake was obtained from the longitudinal flow velocity component on the area of the rotor disk at various downstream positions,

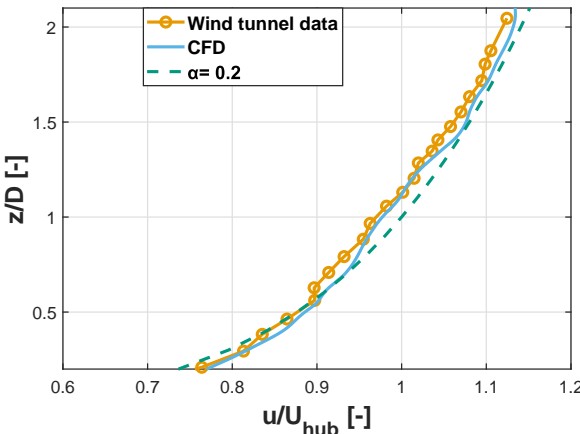

**Figure 3.** Measured and simulated boundary layer profile at the turbine location.

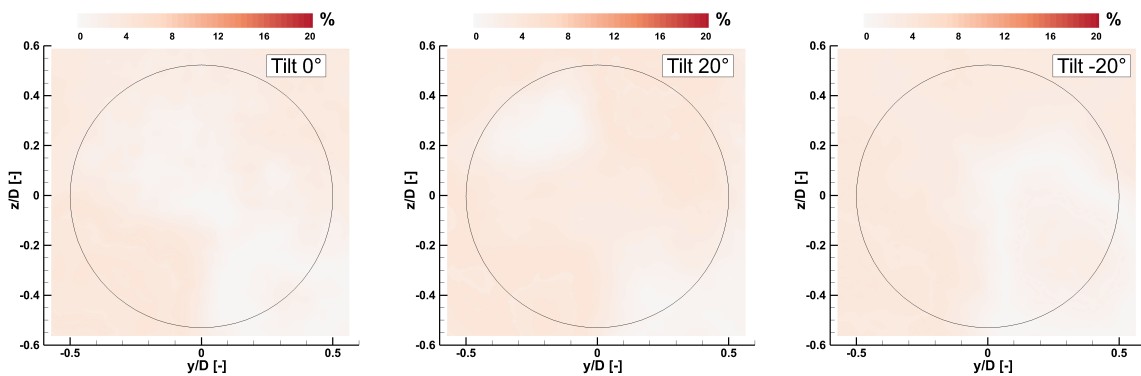

**Figure 4.** Absolute percent error between S-PIV and CFD streamwise flow speed 5D downstream of the rotor, for three tilt angles. The black circle denotes the rotor circumference.

directly behind the wind turbine. Even though, for offshore applications, typical horizontal spacings between two turbines are about equal to 10D, the effects of tilt on downstream power is presented here for distances from 6 to 12D, for the sake of completeness.

Figure 5 shows contours of the normalized streamwise velocity. For brevity, the results are shown only at two downstream distances (namely, $x/D = 6$ and $x/D = 12$) and for three rotor tilt angles ($0°$, $15°$ wake-up and $-15°$ wake-down). The deflection of the wake center is evident at $x/D = 6$, but changes in the flowfield are still noticeable even at $x/D = 12$.

Because of the non-uniformity of the sheared inflow, the wake center is deflected towards the ground by 0.2D even for the aligned rotor case. This effect can also be appreciated in Fig. 6, which shows normalized velocity contours on the $xz$ midplane for the same tilt angles. The flow has a higher momentum above the wake than below it. As a result, turbulent mixing is stronger in the top part of the wake, resulting in a non-symmetrical vertical profile, as already observed in previous studies



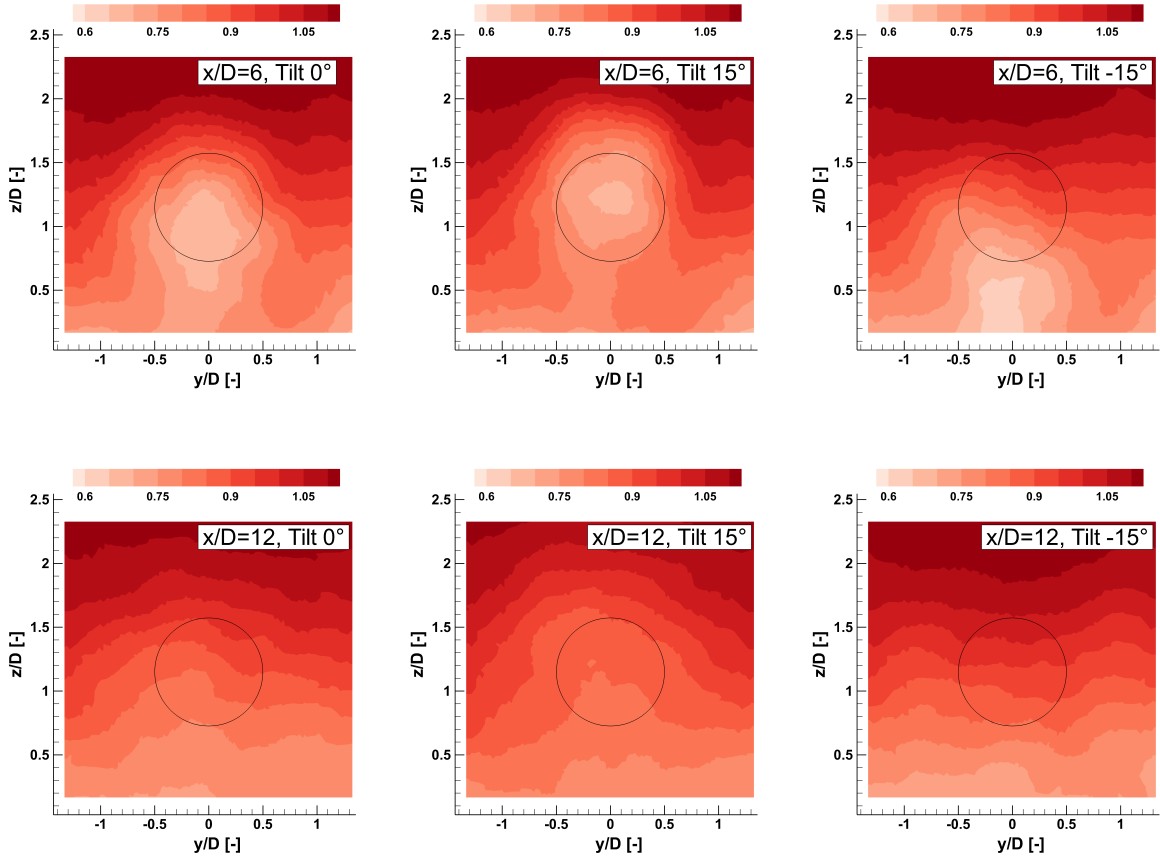

**Figure 5.** Contours of normalized streamwise velocity $u/U_{\text{hub}}$ on $yz$ crossplanes (orthogonal to the flow) for two downstream distances and three rotor tilt angles. The black circle denotes the rotor disk circumference.

(Chamorro and Porté-Agel, 2009; Nanos et al., 2021). Therefore, as shown by the figures, the deflection of the wake towards the sea surface results in a higher energy flow within the downstream rotor area.

Furthermore, it appears that the direction of wake deflection has an effect also on how fast the wake recovers. The recovery rate was calculated based on the integral of the speed $\langle u \rangle$ computed over a square area of size 2.5D-by-2.5D, centered at hub height. This area is sufficiently large to enclose the wake for all simulated tilt misalignment cases. The value of the integral at $x/D = 2$ was considered as a reference, resulting in the following expression of the recovery rate:

$$R_w = \frac{\langle u \rangle}{\langle u \rangle_{2D}}. \tag{2}$$

Figure 7 shows the recovery rate for the tilt angles $0°$, $20°$(wake-up), and $-20°$(wake-down). The plot clearly indicates that the recovery for the wake-up (positive) tilt case is essentially the same of the untilted condition. On the other hand, the recovery rate almost doubles for the wake-down (negative) tilt case.

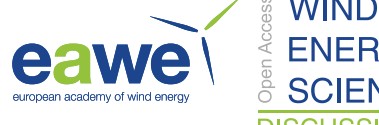

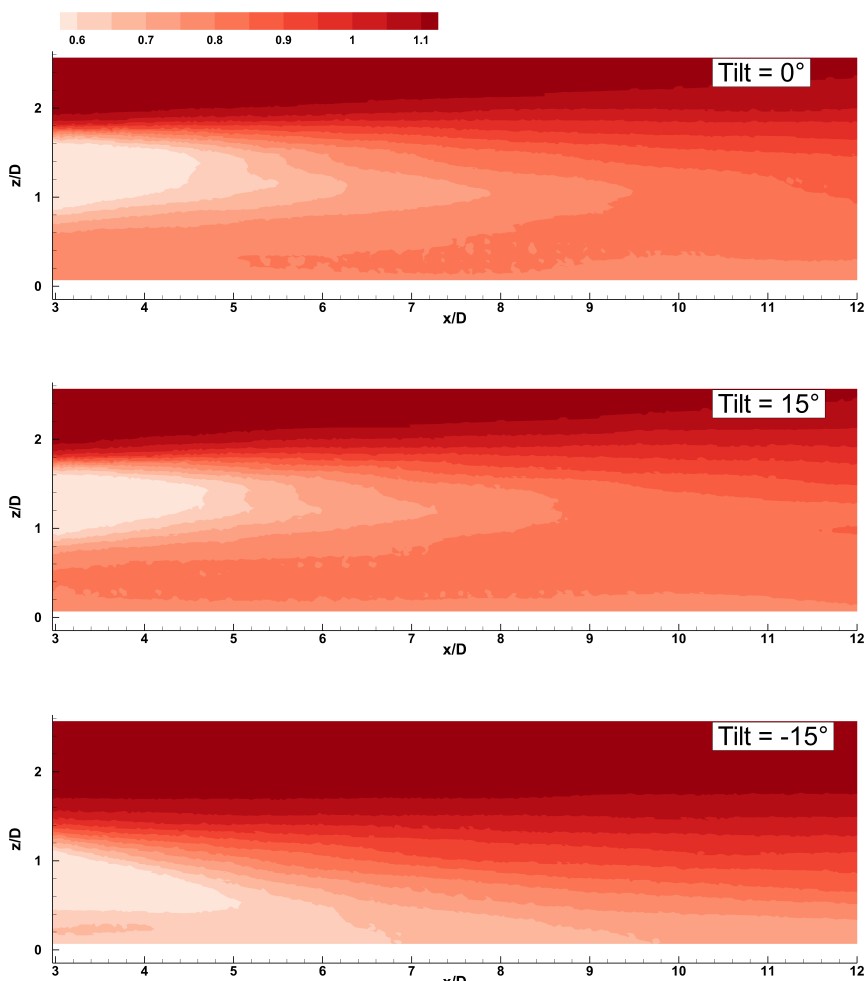

**Figure 6.** Contours of normalized streamwise velocity $u/U_{hub}$ on the $xz$ longitudinal midplane for three rotor tilt angles.

## 3.3 Effects of tilt on power

Figure 8a reports the percent power drop for the wake-deflecting wind turbine as a function of the rotor tilt angle $\beta$. As expected,
tilt misalignment reduces the power capture of the turbine, similarly to the classical yaw misalignment case (Campagnolo et al., 2016b). Best fitting the cosine law $P = P_{\beta=0} \left( \cos \beta \right)^p$ results in an exponent $p \approx 3.5$. For the same scaled turbine, the power drop cosine exponent in yaw misaligned conditions and laminar inflow is $p \approx 2.01$ (Nanos et al., 2021). The range of exponent values reported in the literature for lateral steering is quite wide (Nanos et al., 2021). The value reported here is relatively high, possibly because of the high turbulence intensity and sheared inflow. Vertical shear also contributes to the noticeable lack of
symmetry between wake-up and wake-down tilt angles, in addition to the vertical motion of the hub. The results of the figure





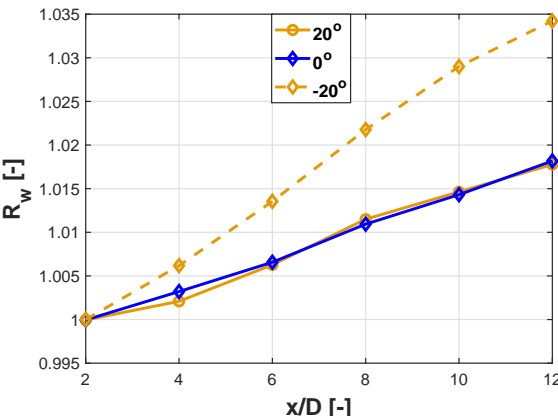

**Figure 7.** Wake recovery rate $R_w$ for $0°$ (untilted), $20°$ (wake-up), and $-20°$ (wake-down) rotor tilt angle.

correspond to the configuration shown in Fig. 2a, where the rotor is facing away from the other two columns; effects caused by the vertical motion depend on the yaw orientation with respect to the platform and on the platform configuration.

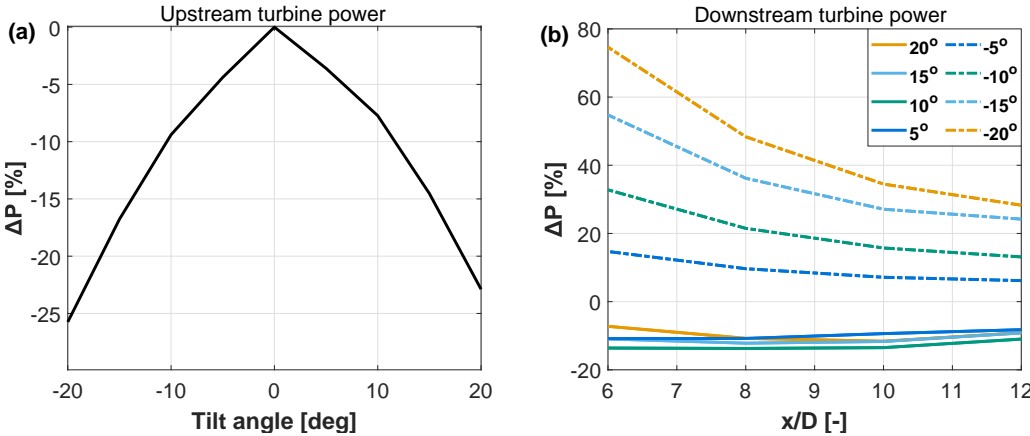

**Figure 8.** Power drop as a function of tilt angle for the wake-steering turbine **(a)**. Power change in the wake as a function of tilt angle and downstream distance **(b)**.

Figure 8b shows the percent power change in the wake as a function of upstream rotor tilt and downstream distance. For wake-down tilt angles, there is a substantial power gain due to the deflection of the wake out of the downstream rotor area, as shown in Fig. 5. The power gain grows with increasing tilt angles, since the wake is further deflected out of the rotor area; at the same time, the upstream machine extracts less power from the flow, resulting in a slightly weaker wake. Furthermore, the power gain decreases with increasing turbine spacing, because of wake recovery.

For wake-up tilt angles, the behavior of the power available in the wake is markedly different, and power drops for all tested tilt angles. In fact, as previously observed with the help of Fig. 6, the upper part of the wake recovers substantially faster than



the lower part. Hence, by pushing the wake upwards, a lower energy flow is moved into the rotor disk area, reducing power capture downstream.

Additionally, it appears that for wake-up tilt angles the power of the second wind turbine is fairly insensitive to the tilt misalignment of the upstream rotor, in contrast to the wake-down case. This effect can be explained with the help of Fig. 9, which shows vertical profiles of the streamwise velocity for various tilt angles at hub center and $x/D = 6$. The effect of
increasing the tilt from $-15°$ to $-20°$ (wake-down) is quite clear, with the larger (negative) tilt angle leading to a higher speed within the rotor disk area. On the other hand, moving from $15°$ to $20°$ (wake-up) has a double-sided effect. On the one hand, the velocity drops in the upper part of the rotor disk area, since the wake center is deflected further up. However, on the other hand the upstream machine extracts less energy from the flow, which results into higher speeds in the lower part of the rotor area. These two effects counteract each other and, as a result, the power production of the downstream machine is relatively
insensitive to the tilt angle for wake-up misalignments.

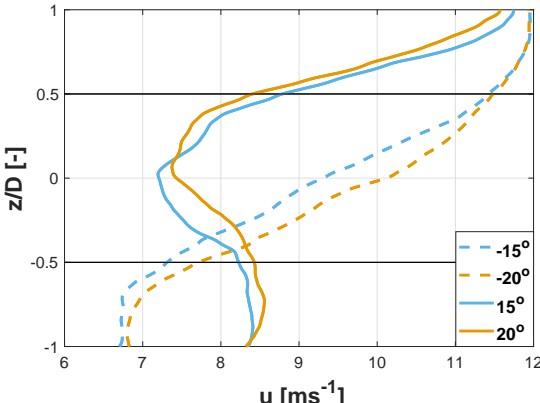

**Figure 9.** Vertical profiles of the longitudinal speed in the wake at $x/D = 6$ for various tilt angles. Black lines denote the rotor upper and lower edges.

Finally, it is interesting to consider the combined gain-loss effects on the power output of the two-turbine cluster, which are shown in Fig. 10. For wake-up tilt angles, the cluster power is always less than the baseline untilted case, which is an expected results since, as shown in Fig. 8, tilt misalignment has a negative impact on both machines. For wake-down tilt angles, the power gain for longitudinal spacings between 10-12D is around 2%, while for closer spacings it can reach up to 7%. The
power gain is small but not negligible, considering the fact that it is observed in a cluster of only two turbines. In fact, previous research has shown that wake deflection strategies can multiply their impact in deep arrays (Annoni et al., 2017; Cossu, 2020a, b).

For a comparison with the more popular method of wind farm control by lateral wake deflection, the same simulation set-up was used to implement lateral —as opposed to vertical— misalignment for three different yaw angles, namely $10°$, $15°$, and
$20°$. The results are reported in Fig. 10 using black markers, and indicate that —for similar misalignment angles— vertical



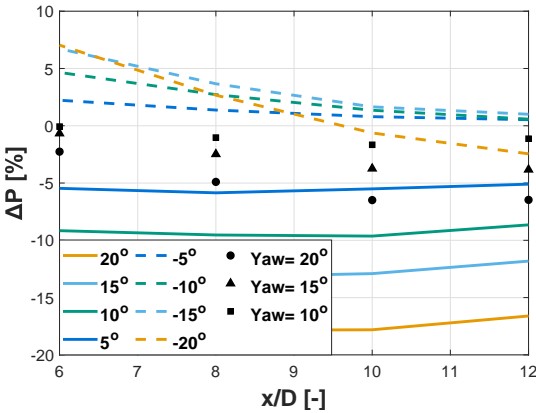

**Figure 10.** Cluster power change as a function of spacing between the turbines, for different tilt misalignment angles (solid and dashed lines). Black markers denote the change in cluster power for different yaw misalignment angles.

deflection towards the ground outperforms lateral deflection. However, this finding is clearly set-up specific, and different layouts might lead to different results. This highlights another important consideration: the two techniques of lateral and vertical steering should not be seen as antagonistic, but rather they could be used together in synergy to achieve the best possible result depending on the layout and operating condition.

## 3.4 Effect of the configuration

The configuration of the floating platform may introduce additional parameters into the problem. For example, with reference to Platform A, pitching results also in a vertical translation of the hub (see Fig. 11). The magnitude of this vertical translation $\Delta z$ depends on the geometric characteristics of the platform-turbine assembly. For the case shown in the figure, the vertical displacement can be computed as

$$\Delta z = \sin\left(\tan^{-1}\left(\frac{1}{m}\right) + \beta\right)\left(\sqrt{l^2 + m^2}\right) - l, \tag{3}$$

where $m$ and $l$ are the horizontal and vertical distances, respectively, between the hub and the center of rotation, and $\beta$ is the pitch angle of the platform (Fig. 11). The center of rotation is considered fixed at the centroid of the platform waterplane area (center of flotation — indicated as CF in the figure), which is a good approximation for small pitch angles (Newman, 2018). The vertical displacement also depends on the turbine orientation with respect to the platform (Fig. 12). For example, consider

the situation of Fig. 11, which corresponds to orientation (a) of Fig. 12. In this case, the rotor center is translated downwards for a wake-down pitching of the platform. However, the opposite happens for orientation (c), where a wake-down pitching of the platform moves the rotor center upwards. Clearly, the ensuing effects on the rotor power and its wake also depend on the amount of shear of the inflow.

Figure 13a shows the relation between rotor tilt angle and the normalized vertical translation of the hub, for the three different

turbine orientations of Fig. 12; these conditions provide an envelope, within which all other possible conditions are contained.



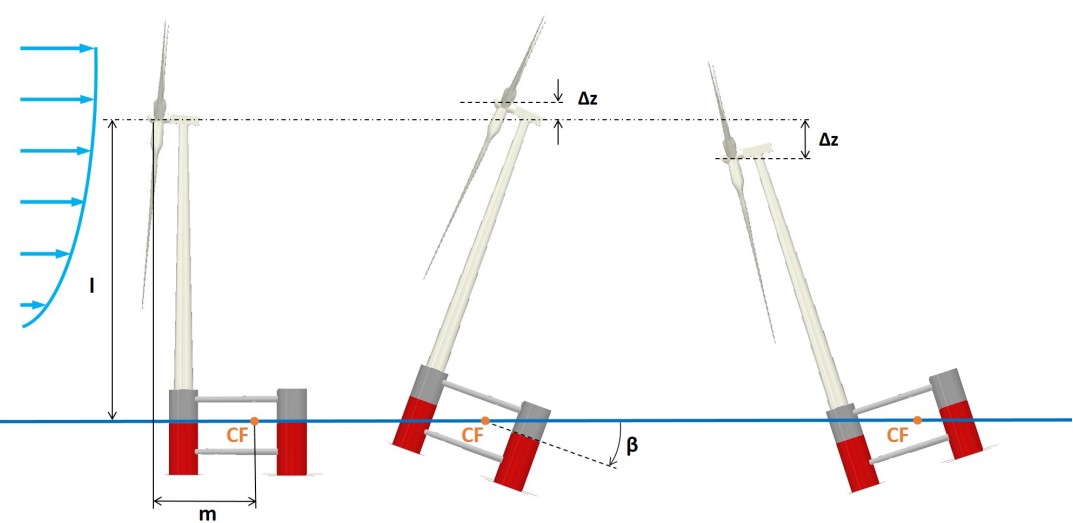

**Figure 11.** Schematic representation of the effects of platform pitch on rotor hub height, considering Platform A. CF: center of flotation; $\beta$: platform pitch angle.

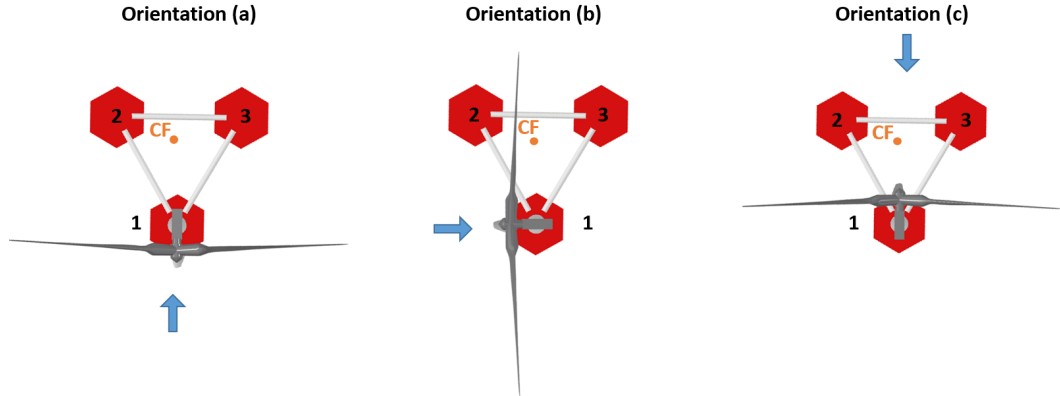

**Figure 12.** Top view of three different turbine orientations with respect to the platform, in the case of Platform A. The blue arrow shows the incoming wind direction. Numbers identify the platform columns.



For Platform B, where the turbine is placed at the center of the floater, this effect is negligible, and all turbine orientations coincide (with a maximum 2% deviation) with orientation (b) of Fig. 12.

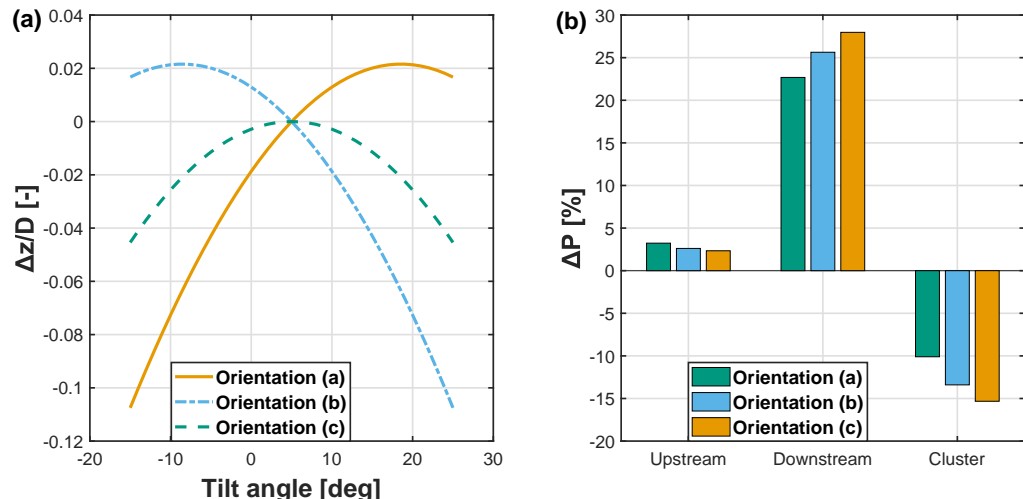

**Figure 13.** Normalized vertical translation of the rotor center as a function of rotor tilt angle for three different turbine orientations **(a)**. Percent power change with respect to the baseline untilted case for a $-15°$ (wake-down) deflection at a 10D downstream distance, for three orientations of the turbine with respect to the floater **(b)**.

The vertical translation of the hub caused by platform pitch can have two effects. First, it slightly shifts the wake position in the vertical direction. Second, in case of a sheared inflow, it exposes the rotor to a slightly faster of slower wind speed. In the current set-up and inflow conditions, it was found that the upstream machine looses approximately 5% of power for a 0.1D vertical movement towards the sea surface following an about $15°$ pitching. This loss of power is added to the one caused by tilt misalignment.

It is clear that these effects are strongly dependent on the turbine-platform configuration (for example, they are nearly absent for the Plaform B case), operating conditions and inflow. Therefore, it is difficult to provide a general assessment of their importance. However, in an effort to assess their potential relevance, it was decided to quantify their boundaries in the worst case scenario of Platform A, where the position of the turbine at the very edge of the floater exacerbates these effects. Figure 13b shows the percent power change with respect to the baseline untilted case for the upstream turbine, the downstream turbine, and the whole cluster. The spacing between the turbines is equal to 10D. As expected, results indicate that the different turbine orientations have opposite effects on the upstream and downstream machine. For example, orientation (b) is better for the downstream machine but worse for the upstream one, when compared to orientation (a). For the cluster power, orientation (c) exhibits the best results with a 3.3% power increase, whereas B and A yield increases of 2.6% and 2.3%, respectively. These results apply to a sheared inflow exponent of 0.2, and would be correspondingly reduced/amplified by less/more pronounced shears.



# 4   Assessment of fatigue and ultimate loads

The feasibility of the proposed vertical wake deflection technique was verified with respect to its effects on structural loading. The goal here is not to precisely characterize the loading of the pitched configurations, but rather to reveal potential unrealistic load increases on the principal structural elements, which cannot be accommodated through confined design modifications.

## 4.1   Simulation set-up

Structural loads were calculated with the comprehensive hydro-aero-servo-elastic analysis tool hGAST (Manolas et al., 2015; Manolas, 2015; Manolas et al., 2020) for the 10 MW reference turbine mounted on the tri-spar concrete floater (Platform B). Mooring lines were modelled using non-linear truss elements, without modifications with respect to the original mooring system of the floater. Load analyses were performed for the baseline untilted configuration, and with the platform pitched by $20°$ in both wake-up and wake-down directions. The analysis considered medium sea-severity conditions, characterized by a 50-year significant wave height of 10.9 m, a peak period of 14.8 s, and a water depth of 180 m; typical offshore wind conditions were considered, characterized by high mean speeds and moderate turbulence levels, corresponding to wind class IB.

Simulations were conducted for a subset of the most critical design load cases (DLCs) of the IEC 61400-3 standard (IEC, 2008), including both extreme and fatigue conditions. The reduced test matrix is reported in Table 3.

**Table 3.** Definition of DLCs for the time-domain hydro-aero-servo-elastic analysis.

| DLC | Wind | Wave | Bins (ms$^{-1}$) | Yaw (deg) | Wave (deg) | Safety factor |
|-----|------|------|------------------|-----------|------------|---------------|
| 1.2 | NTM  | NSS  | 3-25, step 2     | 0         | 0          | -             |
| 1.3 | ETM  | NSS  | 11-25, step 2    | 8         | 0          | 1.35          |
| 1.6 | NTM  | ESS  | 11-25, step 2    | 8         | 0          | 1.35          |
| 6.1 | EWM  | SSS  | 50               | 0, 8      | 0, 30      | 1.35          |
| 6.2 | EWM  | SSS  | 50               | +/- 30    | +/- 30     | 1.10          |

A list of power production (normal operation) cases covering a wide operational range of wind and wave conditions are considered in the 1.x-series. In the table, NTM and ETM refer to the normal and extreme turbulence models, respectively, while NSS and ESS refer to the normal and extreme sea states, respectively. Since wake steering is used only in the partial load region, power production simulations for the pitched platform case are performed for wind speeds up to 13 ms$^{-1}$, to include the next speed bin just above rated (which is equal to 11.4 ms$^{-1}$ for this turbine).

DLC 1.2 corresponds to normal operation of the floating turbine in normal turbulence and sea state, and it is used for estimating the fatigue limit state (FLS). DLCs 1.3 and 1.6 correspond to extreme wind/wave conditions, and are used to estimate the ultimate limit state (ULS). Clearly, there is no actual benefit in energy production from keeping wake steering control in operation when extreme wave conditions are anticipated or encountered (ESS conditions of DLC 1.6). Therefore, it is reasonable to assume that ballast control for wake steering is shut down in these conditions. Weather forecast and sensors (e.g. accelerometers, buoys, etc.) could be used to identify such conditions. However, since the response time of the control





system is relatively slow (in the order of minutes, as indicated later in Sect. 5), DLC 1.6 was retained in the analyses to assess
the effects of a system failure to timely set back the platform to its reference position.

DLC 6.x corresponds to operation under storm conditions, during which the turbine is in idling mode (combined with a
grid loss in DLC 6.2). Typically, as a result of the loss of the grid, yaw control is disabled, possibly leading to extreme yaw
misalignment angles. In many circumstances, such large angles drive the maximum loads on the rotor and the turbine. In the
present analysis, misalignments of 0° and 8° in the wind direction and 0° and 30° in the wave direction were considered in DLC
6.1 (normal idling operation), while +/-30° wind misalignment and co-directional waves were considered in DLC 6.2 (idling
operation combined with grid loss). It is noted that independent studies of an onshore version of the DTU 10 MW turbine have
shown that yaw angles of +/-30° are the most critical in terms of maximum loads (Wang et al., 2017).

Before conducting the time domain simulations, the hydrostatic stability of the tilted floating turbine was confirmed. Further-
more, a modal analysis of the overall system revealed that pitching has extremely limited changes on the natural frequencies,
as shown in Table 4.

**Table 4.** Eigenvalues (Hz) of the coupled floating system for three platform pitch angles ($0°$, $-20°$, and $20°$).

|   | Mode | Pitch=0° | Pitch=−20° | Pitch=20° |
|---|---|---|---|---|
| 1 | Floater Surge | 0.006 | 0.006 | 0.006 |
| 2 | Floater Sway | 0.006 | 0.006 | 0.006 |
| 3 | Floater Yaw | 0.014 | 0.014 | 0.014 |
| 4 | Floater Pitch | 0.043 | 0.047 | 0.046 |
| 5 | Floater Roll | 0.043 | 0.045 | 0.047 |
| 6 | Floater Heave | 0.060 | 0.061 | 0.061 |
| 7 | 1st Tower Side-side | 0.383 | 0.375 | 0.381 |
| 8 | 1st Tower Fore-aft | 0.400 | 0.398 | 0.391 |
| 9 | 1st Rotor Edgewise Symmetric | 0.541 | 0.539 | 0.540 |
| 10 | 1st Rotor Flapwise Yaw | 0.562 | 0.560 | 0.561 |
| 11 | 1st Rotor Flapwise Tilt | 0.600 | 0.599 | 0.598 |
| 12 | 1st Rotor Flapwise Symmetric | 0.646 | 0.646 | 0.646 |
| 13 | 1st Rotor Edgewise Vertical | 0.928 | 0.928 | 0.928 |
| 14 | 1st Rotor Edgewise Horizontal | 0.941 | 0.940 | 0.941 |

The first-order hydrodynamic operators were re-calculated for both tilted floaters. The hydrodynamic problem considers the
floater interacting with the incoming waves, and it is modelled using the hybrid integral equation method freFLOW (Manolas,
2015), which solves the 3D Laplace equation using the Boundary Element Method in the frequency domain. The solution
procedure provides the exciting loads, the added masses and damping coefficients, the response amplitude operators (RAOs)
of the floater, the hydrodynamic pressure on the wet surface of the floater, as well as the linearized hydrostatic stiffness matrix
taking into consideration the exact "mean" geometry of the floater. Comparisons of the surge ($F_{\text{surge}}$) and heave ($F_{\text{heave}}$) exciting





force and the pitch exciting moment ($M_{\text{pitch}}$) are shown in Fig. 14a-c. Results are normalized by gravity $g$, water density $\rho$, and wave amplitude $A$. The figure shows that the tilting of the floater (in either direction) has a relatively small effect on wave excitation loads. Higher localized differences (in the frequency range 0.5-1.5 rads$^{-1}$) are noted in the pitching moment and the

heaving force. Moreover, the pitching of the floater in both directions increases the heave wave exciting loads, by the pressure loads that are generated over the inclined cylindrical surfaces of the columns.

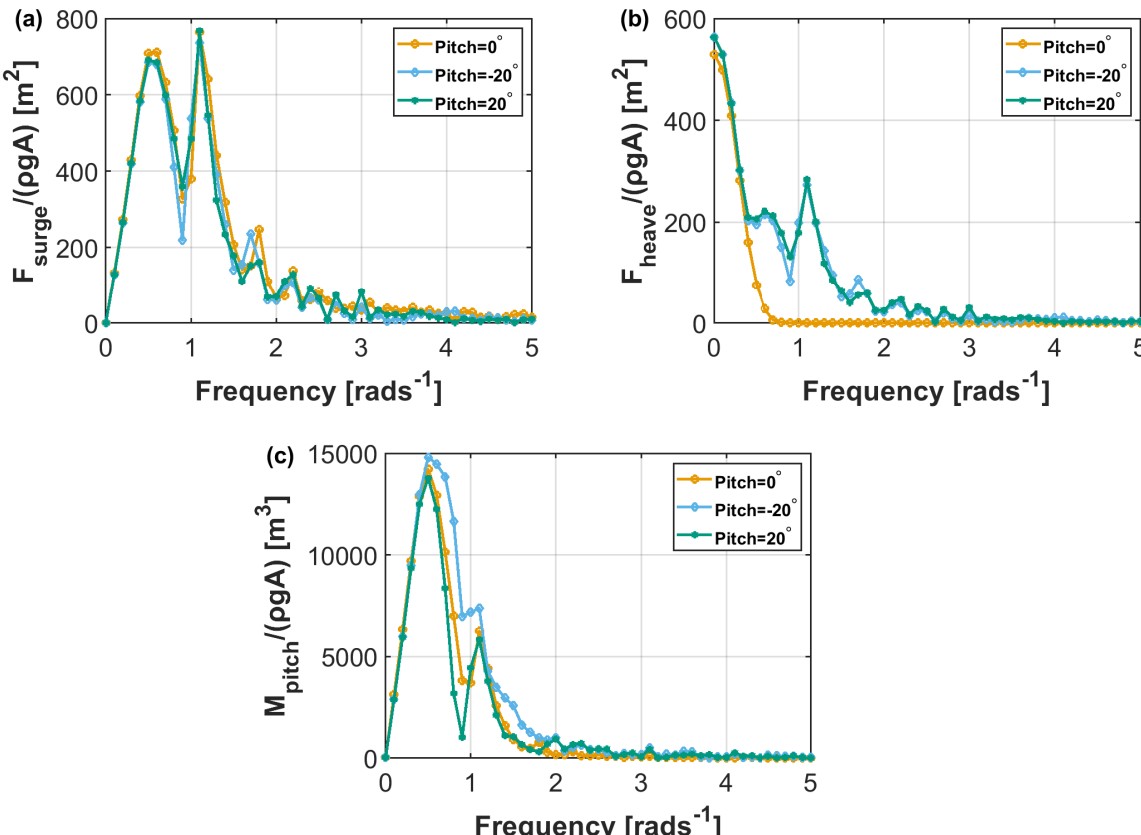

**Figure 14.** Comparison of the wave exciting loads for platform pitch angles $0°$, $20°$ (wake-up), and $-20°$ (wake-down). Surge **(a)**; heave **(b)**; pitch **(c)**.

## 4.2 Effects on damage equivalent and ultimate loads

Table 5 reports lifetime damage equivalent loads (DELs) of the two tilted configurations, comparing them with the ones of the untilted baseline case. DEL calculations are solely based on load time series from DLC 1.2, neglecting parked conditions

and startup/shutdown sequences. Lifetime DELs are obtained considering a typical Weibull distribution with $C = 11.3$ ms$^{-1}$ and $k = 2$, $10^7$ reference cycles and a lifetime of 20 years. The S-N curve exponents $m = 4$, 8, and 10 are used for the tower/mooring lines, drive-train, and blades, respectively. Results indicate that blade, drive-train and tower FLS loads are



barely affected by the tilting of the turbine in either direction. Overall, there is a slight reduction in the edgewise loads, which
is higher (5%) for wake-up pitching. This is explained by the fact that the component of the weight load that lies over the rotor
plane decreases as the turbine is tilted. The asymmetry between the wake-down and wake-up case is due to the built-in 5° uptilt
angle of the rotor, which increases the overall rotor tilt for wake-up inclination of the turbine and visa-versa. Flapwise bending
moments remain unaffected. Tower bending moments slightly increase for wake-down pitching, whereas they slightly decrease
for wake-up pitching. A maximum increase of 2% is noted on tower base moments. Tower yaw moments slightly decrease in
both configurations.

**Table 5.** Comparison of lifetime DELs for platform pitch angles $0°$, $20°$ (wake-up), and $-20°$ (wake-down).

| Sensor | Units | Pitch=$0°$ | Pitch=$-20°$ | Pitch=$20°$ |
|---|---|---|---|---|
| Blade root edgewise moment | kNm | 28 503 | −1% | −5% |
| Blade root flapwise moment | kNm | 31 253 | 0% | 0% |
| Blade root torsion moment | kNm | 454 | 2% | −2% |
| Drivetrain yaw moment | kNm | 28 878 | 0% | 0% |
| Drivetrain tilt moment | kNm | 28 678 | 0% | 0% |
| Drivetrain torsion moment | kNm | 3 356 | 5% | 0% |
| Tower base side-side moment | kNm | 77 492 | 2% | −2% |
| Tower base fore-aft moment | kNm | 272 255 | 2% | −6% |
| Tower base yaw moment | kNm | 27 160 | −1% | −2% |
| Tension at fairleads | kN | 176 | 13% | 1% |
| Tension at anchors | kN | 169 | 12% | 1% |

The main effect of pitching the platform is seen on the mooring line loads, specifically in the case of wake-down pitch. An
increase of 13% and 12% is noted on the tension load at fairleads and at the anchor positions, respectively. On the other hand,
only a minor increase of 1% is observed for the wake-up pitch case. DEL estimates are derived by averaging over the three
mooring lines.

    Regarding ultimate loads, it was found that in all cases loads were driven by DLCs outside of the operational envelope of
the ballast control system (i.e., DLC 1.6 or DLC 6.x corresponding to extreme sea state or parked/idling operation, or at wind
speeds higher than the wake-steering cut-off speed of 13 ms$^{-1}$). Even though the ballast control system will not be used in
extreme sea states, it was decided to include DLC 1.6 in the ultimate load analysis with the aim of assessing the effect on loads
of the platform remaining pitched under extreme wave conditions.

    Results are summarized in Table 6, which also reports the originating DLCs. Changes in loads are null for all cases where
ULS are found in DLCs other than 1.6 or 1.3 up to 13 ms$^{-1}$, because the ballast control system is assumed to be deactivated
in such conditions. When ultimate loads are originated in DLC 1.6, the load change obtained when this DLC is excluded is
indicated in parenthesis.



**Table 6.** Comparison of ultimate loads for platform pitch angle $0°$, $20°$, (wake-up) and $-20°$ (wake-down).

| Sensor | Units | Pitch=$0°$ | | Pitch=$-20°$ | | Pitch=$20°$ | |
|---|---|---|---|---|---|---|---|
| | | ULS | DLC | ULC | DLC | ULC | DLC |
| Blade root edgewise moment | kNm | 32 717 | $6.2-50$ ms$^{-1}$ | 0% | $6.2-50$ ms$^{-1}$ | 0% | $6.2-50$ ms$^{-1}$ |
| Blade root flapwise moment | kNm | 83 212 | $1.6-13$ ms$^{-1}$ | 5% (0%) | $1.6-13$ ms$^{-1}$ | 5% (0%) | $1.6-13$ ms$^{-1}$ |
| Blade root torsion moment | kNm | 1 309 | $6.2-50$ ms$^{-1}$ | 0% | $6.2-50$ ms$^{-1}$ | 0% | $6.2-50$ ms$^{-1}$ |
| Blade root combined moment | kNm | 83 541 | $1.6-13$ ms$^{-1}$ | 5% (0%) | $1.6-13$ ms$^{-1}$ | 6% (0%) | $1.6-13$ ms$^{-1}$ |
| Drivetrain yaw moment | kNm | 52 995 | $1.3-25$ ms$^{-1}$ | 0% | $1.3-25$ ms$^{-1}$ | 0% | $1.3-25$ ms$^{-1}$ |
| Drivetrain tilt moment | kNm | 55 680 | $1.6-25$ ms$^{-1}$ | 0% | $1.6-25$ ms$^{-1}$ | 0% | $1.6-25$ ms$^{-1}$ |
| Drivetrain torsion moment | kNm | 17 227 | $1.3-25$ ms$^{-1}$ | 0% | $1.3-25$ ms$^{-1}$ | 0% | $1.3-25$ ms$^{-1}$ |
| Drivetrain combined moment | kNm | 63 406 | $1.3-25$ ms$^{-1}$ | 0% | $1.3-25$ ms$^{-1}$ | 0% | $1.3-25$ ms$^{-1}$ |
| Tower base side-side moment | kNm | 507 594 | $6.2-50$ ms$^{-1}$ | 0% | $6.2-50$ ms$^{-1}$ | 0% | $6.2-50$ ms$^{-1}$ |
| Tower base fore-aft moment | kNm | 808 458 | $1.6-13$ ms$^{-1}$ | 19% (0%) | $1.6-13$ ms$^{-1}$ | 39% (0%) | $1.6-11$ ms$^{-1}$ |
| Tower base yaw moment | kNm | 55 476 | $1.3-25$ ms$^{-1}$ | 0% | $1.3-25$ ms$^{-1}$ | 0% | $1.3-25$ ms$^{-1}$ |
| Tower base combined moment | kNm | 808 458 | $1.6-13$ ms$^{-1}$ | 19% (0%) | $1.6-13$ ms$^{-1}$ | 40% (0%) | $1.6-11$ ms$^{-1}$ |
| Tension at fairleads | kN | 6 403 | $1.6-11$ ms$^{-1}$ | 7% (0%) | $1.6-11$ ms$^{-1}$ | $-2$% (0%) | $6.1-50$ ms$^{-1}$ |
| Tension at anchors | kN | 5 965 | $6.1-50$ ms$^{-1}$ | 7% (0%) | $1.6-11$ ms$^{-1}$ | 0% | $6.1-50$ ms$^{-1}$ |

Results indicate that blade extreme loads are only slightly affected by the platform pitching (maximum increase of 5-6%), while drivetrain loads remain unaffected. Larger increases in ultimate loads are noted on tower bending moments. A 19%
increase in the combined bending moment is seen in the case of wake-down pitching, whereas a substantially higher increase (40%) is seen in the case of wake-up pitching. The driving load case for tower base loads is DLC 1.6 around rated wind speed, both for the baseline and the pitched configurations. The difference between wake-down and wake-up pitch for tower base fore-aft moment is due to the direction of thrust. For both pitch angles, the moment arm of the weight of the rotor nacelle assembly (RNA) increases, with a corresponding increase of the bending moment at the foot of the tower. However, the fore-aft
moment due to the rotor thrust adds to the moment caused by the RNA weight for the wake-up pitch case, whereas it reduces the moment in the wake-down pitch case.

The tension of the mooring lines increases when the platform is pitched wake-down (by 7%, caused by DLC 1.6), whereas slightly decreases (by 2%) in case of wake-up pitching. In the latter case, the tension loads in the mooring lines obtained in DLC 1.6 are smaller than those obtained in DLC 6.1, which becomes the driving load case.
Overall the above results indicate that it is beneficial to deactivate the ballast system when extreme wave conditions occur. This is especially true for the tower base bending moment, which experiences the highest increase. It is worth noting, however, that the observed 40% load increase comes from the wake-up platform attitude, which, as shown in §3.3, is not capable of boosting power output. For the more interesting wake-down pitch case, the increase in tower base combined moment is much



more contained. Most importantly, based on the reasonable assumption that platform pitching is deactivated for DLC 1.6

conditions, it appears that ballast control will not increase the ultimate loads on the structure.

## 5 Ballast movement estimate

Next, the hydrostatics of floating bodies was used to estimate the differential ballast control necessary to pitch the two floating turbine configurations considered here. For a platform that is resting horizontally, all forces and moments are in equilibrium. If ballast is moved in a specific direction, the center of gravity will move accordingly. As a consequence, the platform will

pitch moving the center of buoyancy in the same direction, until a new equilibrium condition is reached (Patel, 1989). The calculations presented here are only indicative and, here again, specific to the configurations considered. For instance, some assumptions have to be made regarding the water ballast pumping system. Moreover, the orientation of the turbine with respect to the floater also plays a role for Platform A, since the ballast movement required for orientation (b) is different from the one necessary for orientations (a) and (c) (see Fig. 12).

The analysis is performed for a rotor tilt angle of $-15°$ (wake-down), which is associated with significant power gains. Given the $5°$ uptilt of the DTU 10 MW turbine, the platform is pitched forward by $20°$ to achieve this attitude. Based on the findings of §3.4, for Platform A the analysis is conducted for orientation (c), which is the most beneficial case in terms of cluster power gain. As previously noted, due to the more symmetric configuration of Platform B, the turbine-platform orientation does not play a role in that case.

For Platform B, the necessary ballast movement was found from static simulations in horizontal and pitched attitudes using the hGAST software (Manolas et al., 2015; Manolas, 2015; Manolas et al., 2020).

Since a similar structural model of Platform A was not available, in this case ballast movement was estimated based on simpler equilibrium of forces and moments for the horizontal and pitched configurations. The analysis considered the mass distribution of the platform and of the turbine, the turbine thrust and torque, and the buoyancy forces from the three columns

(Fig. 15). The forces transmitted to the platform by the mooring lines were neglected from the analysis, because the pitch/roll mooring stiffness is small for catenary lines compared to the hydrostatic and gravitational stiffness. This assumption was validated using hGAST for Platform B, where the mooring lines were found to contribute only 5% of the total restoring force. Since no detailed design for Platform A was available, the precise displacement of the column center of gravity and buoyancy could only be estimated. However, the sensitivity of the results to these quantities is relatively low. For example, it was verified

that, even with a 30% deviation from the estimated values (which is an exaggerated assumption), the final results of the ballast calculations are affected by less than 4%. To verify the calculation method used for Platform A, the same approach was used also for Platform B, and the results were compared to the ones obtained with hGAST software, yielding only a 6% difference. Such accuracy was deemed sufficient for the preliminary nature of the present investigation.

Results indicate that the necessary ballast movement to achieve a $-20°$ wake-down pitch attitude for Platform A is approx-

imately equal to 500 m$^3$. Considering configuration (c) (Fig. 12c and 15), assuming that water can be moved between each column by dedicated 60 kW pumps, two pumps are used to move water in order to lift column 1 and sink columns 2 and 3. As





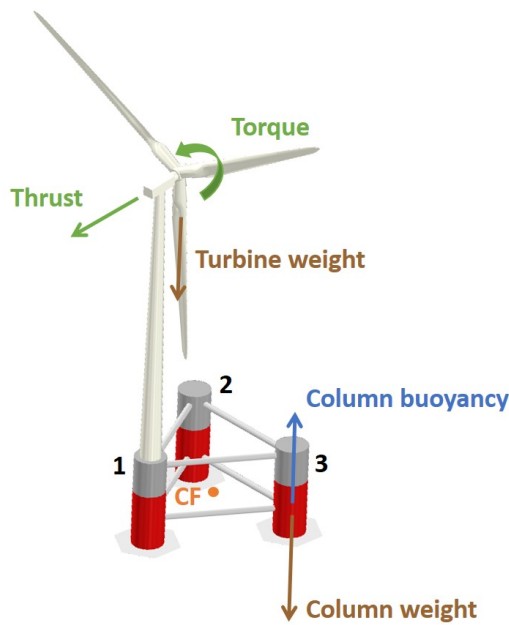

**Figure 15.** Forces and moments considered for the ballast calculations. For clarity, only the forces on one of the columns are shown.

the platform pitches towards a $-20°$ attitude, a height difference between column 1 and columns 2-3 is created. This means that the ballast moves towards a lower position, which facilitates the maneuver. This change of attitude necessitates of approximately 5 minutes and 2 kWh (Menon, 2004). When it is time to return the platform to the horizontal attitude, the situation

is different, since the ballast has to move to a higher position. In this case, the maneuver takes approximately 13 minutes and costs roughly 26 kWh.

For the same $-20°$ wake-down pitch attitude, the ballast required for Platform B is equal to $2\,900$ m$^3$. Given the very large size and weight of this configuration, assuming pumps of triple the power than for case A, the first maneuver (from horizontal position to $-20°$ pitch attitude) takes around 10 minutes, and the energy expenditure is of about 13 kWh. The return to a

horizontal attitude requires slightly more than 20 minutes and costs 132 kWh.

Considering two reference turbines spaced 10 rotor diameters apart, exposed to an ambient wind speed of 9 ms$^{-1}$, the front turbine produces 5 MW and the downstream machine yields 3.5 MW. In such a condition, tilting the first turbine would improve the cluster power production by roughly $3\%$, i.e. 250 kW (which is a conservative assumption, given the results of §3.4). Given that the relation between tilt angle and power gain is approximately linear, Platform A and B would respectively need about 13

and 51 minutes of tilted operation (including the transition time from horizontal to target tilt angle) to break even the energy expenditure caused by tilting, and start having a net energy gain.



As previously mentioned, the orientation of the turbine with respect to the platform plays a role for Platform A. In fact, the ballast movement required for orientation (b) is three times larger than for orientations (a) and (c); this effect is however negligible for Platform B.

Notwithstanding the variety of possibilities and the room for further optimization, these results indicate that tilt control by differential ballast is a rather slow control input that should be activated in fairly steady wind conditions; possibly, faster changes in ambient conditions could be tracked by lateral yaw misalignment. Additionally, the characteristics of the platform also play a role, with heavy configurations being at a significant disadvantage.

## 6   Conclusions

This paper has presented a technical feasibility assessment of vertical wake steering for floating wind turbines. Today the most mature wind farm control approach is lateral wake steering, a method that is attracting significant attention as the wind energy community is trying to alleviate the adverse effects of turbine wake interaction within wind plants. One of the reasons behind the success of lateral deflection is the fact that it can be implemented without a radical redesign of the turbine. The present study is an attempt at verifying if a similar approach is possible also for vertical steering in floating turbines.

The study is based on two different floating platforms and one reference 10 MW wind turbine. These platforms feature ballast tanks for balancing the structure, and incorporate an active ballast control system for keeping the platform aligned with the water surface (Roddier et al., 1997). The idea explored here is to reuse or adapt such systems in order to tilt the rotor and deflect the wake vertically.

    Results obtained with a combined simulation-experimental study indicate that, for two aligned wind turbines spaced 10-12

diameters apart, power gains reach about 3%, while for spacings of 6-8 diameters gains can increase to about 7%, similarly to the findings of previous research (Annoni et al., 2017; Cossu, 2020a; Bay et al., 2019). These gains are obtained with a $-15°$ wake-down rotor tilt angle that corresponds, due to the rotor uptilt, to a $-20°$ pitch forward of the platform. Because of the direction of rotor uptilt, smaller platform rotations would be necessary for downwind turbine configurations. Nowithstanding this possible advantage, downwind turbines were not considered in this work because they are effectively absent from the cur-

rent market. However, they have other interesting characteristics for very large rotors that might possibly change this situation in the future (Loth et al., 2017).

    The present study has only considered two turbines in full waked conditions, for a specific ambient shear and turbulence intensity. However, previous research has shown that power gains may further increase when considering a larger number of turbines and more complex configurations (Annoni et al., 2017; Cossu, 2020a; Bay et al., 2019). In accordance with prior

studies on vertical wake steering (Fleming et al., 2015; Annoni et al., 2017; Cossu, 2020a), the present results confirm that deflecting the wake towards the sea surface is more effective than deflecting it towards the sky. In fact, wake recovery is not axisymmetric when the wake develops within a boundary layer. Due to the vertical non-uniformity of the free stream, turbulent mixing and recovery are faster in the top than in the bottom part of the wake. Therefore, deflecting the wake towards the sea surface results into an air flow of higher momentum moving downwards and into the downstream rotor disk area,



thereby boosting capture; the opposite happens when the wake is deflected upwards, resulting in a slower flow being lifted up towards the downstream rotor. Additional intra-plant phenomena happen when considering larger arrays and more complex configurations (Cossu, 2020a, b), further increasing power capture.

Another conclusion of the present study is that the geometric characteristics of the platform can have a substantial effect. According to intuition, it was found that a steel platform with smaller draft requires much less ballast movement compared

to a concrete platform with greater draft. Specifically, the lighter-weight three-floater configuration of Platform A is able to transition from a horizontal no-steering condition to full steering in about 5 minutes. For a two-turbine cluster, it would take about 13 minutes from the beginning of the maneuver to compensate the expenditure due to tilting and start gaining power; on deeper arrays (Cossu, 2020a, b) this time might be substantially reduced, because of the larger power gains. On the other hand, longer maneuvering times and higher energy expenditures penalize heavy large-draft configurations as the one represented by

Platform B. Ballast movement however also depends on additional details related to the geometry of the system. For example, the orientation of the turbine with respect to the platform can have an effect on ballast when the turbine is located directly above a column, because of the strong inertial asymmetry that it creates. Additional effects are related to vertical movements of the hub caused by pitching, resulting in small changes in power capture for sheared flow conditions, which can be beneficial or detrimental depending on the direction of the vertical motion. A more comprehensive analysis, reflecting the latest and

most promising configurations, is necessary before final conclusions can be drawn on whether ballast movement is a viable option for implementing vertical steering by pitching. However, this initial study seems to indicate that the amount of water that needs to be moved, the time it will take to pitch and the energy that is required, are not unrealistically high, at least for the lighter-weight Platform A configuration. The present results also indicate that a light-weight steel configuration (like Platform A) with a central arrangement of the turbine (like Platform B) might seem to offer an interesting solution, worth investigating

in future studies.

Even in the most beneficial conditions, this preliminary analysis clearly shows that rotor tilting by differential ballast control is a relatively slow control input. Therefore, vertical steering by this method should probably be used only to follow slow changes in wind conditions. On the other hand, lateral steering is able to operate on somewhat faster time scales. This seems to give a strong suggestion towards the study of integrated lateral-vertical steering control, which should try to combine these

two complementary methods to maximize their synergies. Of course, steering is only one of the various wake manipulations techniques available, and it could be integrated with —for example— induction control. The optimal combination of techniques for affecting wake behavior is an active area of research in wind farm control, and further progress is anticipated.

Another key aspect related to the feasibility of the present wind farm control method is related to the loading experienced by the steering turbine. This problem was investigated by hydro-aero-servo-elastic simulations with reference to the Platform

B concept, for which a complete structural model was available. This heavy platform with a large draft does not seem to be ideally suited to vertical steering, because of the large ballast movements that it requires. In hindsight, a loading study of the lighter weight Platform A would have been more appropriate; this was unfortunately not possible within the scope of the present project, because a detailed design of the Platform A concept was not available. Loads were evaluated for 20° pitch forward and pitch backward attitudes, considering both fatigue and ultimate loads, and they were compared to the design loads





of the floating system when it is operated without vertical wake steering. The comparison was made under the assumption that steering is used only up to speeds just above rated, and that it would not be used in extreme sea and wind conditions. A detection system with appropriate redundancy might possibly be used to ensure safe behavior in operation, although the characteristics of such a system were not considered in this work. Results indicate that there is only a minor effect on turbine fatigue loads, with an increase of about $5\%$ being experienced by the drivetrain torsional moment. On the other hand, larger

increases of about 12-13% were noted on the mooring system, which would have to be accordingly redesigned. Ultimate loads were not affected, since —for this turbine and platform— they are all produced in operational conditions where wake steering is not utilized. These results are promising, but here again more specific analyses are needed before more conclusive answers can be given.

*Data availability.*   Data from the CFD and hydro-aero-servo-elastic simulations is available upon request.

*Author contributions.*   EMN conducted the experiments, contributed to the CFD simulations, performed all sizing analyses and contributed to the interpretation of the results; CLB devised the original idea of vertical wake steering by differential ballast control, collaborated in the interpretation of the results and supervised the overall work; DIM and VAR performed the hydro-aero-servo-elastic analysis, and contributed to the analysis of the loading conditions on the turbine. EMN and CLB wrote the manuscript, except for Sect. 4 that was primarily authored by DIM and VAR. All authors provided important input to this research work through discussions, feedback and by improving the manuscript.

*Competing interests.*   The authors declare that they have no conflict of interest.

*Acknowledgements.*   The authors would like to thank Helena Canet at TUM for her input on the load analysis, and Dr. Antonis Daskalakis of Offshore Energy Systems SA for his advice on the platform sizing. Chengyu Wang at TUM contributed to the work on the CFD simulations, while Giacoma Valerio Iungo, Kyle Jones, and Mario Rotea, all at UTD, supported the wind tunnel measurements at UTD.





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

## Nomenclature

| | |
|---|---|
| $A$ | Wave amplitude |
| 625   $C_P$ | Power coefficient |
| $D$ | Rotor diameter |
| $F$ | Force |





| | | |
|---|---|---|
| | $M$ | Moment |
| | $g$ | Gravitational acceleration |
| 630 | $p$ | Cosine law power loss exponent |
| | $P$ | Power |
| | $R_w$ | Wake recovery rate |
| | $U$ | Ambient wind speed (time averaged) |
| | $u$ | Streamwise velocity component (time averaged) |
| 635 | $x$ | Streamwise coordinate (positive downstream) |
| | $y$ | Crosswind coordinate (positive left, looking downstream) |
| | $z$ | Vertical coordinate (positive up) |
| | $\beta$ | Platform pitch angle |
| | $\rho$ | Water density |
| 640 | ALM | Actuator-line method |
| | CFD | Computational fluid dynamics |
| | CF | Center of flotation |
| | DEL | Damage equivalent load |
| | DLC | Design load case |
| 645 | ETM | Extreme turbulence model |
| | EWM | Extreme wind model |
| | FLS | Fatigue limit state |
| | LES | Large-eddy simulation |
| | NSS | Normal sea state |
| 650 | NTM | Normal turbulence model |
| | S-PIV | Stereo-Particle image velocimetry |
| | SSS | Severe sea state |
| | ULS | Ultimate limit state |