# Peer review of "Vertical wake deflection for floating wind turbines by differential ballast control"

_Wind Energy Science, 2021_

## Author Comment (AC1)

**Reply to Reviewers**

The authors would like to thank the two reviewers for their time and for the useful feedback. All inputs that they provided have contributed to the improvement of the paper.

A list of point-by-point replies to the reviewers' comments is reported in the following.

**Reviewer #1**

1. **Reviewer:** *In figure 4 the percent error is shown. Is this a time average error or is the data from a single time shot. It might also be nice to use a different scale, the main percentual range looks between 0-10%. That way the differences might become more apparent.*
   **Authors:** It is the time-average error. The text has been modified to clarify this point. We also changed the scale to 0-6%, as suggested.

2. **Reviewer:** *For equation 2, I find it difficult to understand the "rate" aspect of this equation. As it is defined it shows how much "more" wind there is at each distance downstream, but I wouldn't say it is a measure of how quickly the wake is recovering, only by how much it has recovered. As it is now the word "rate" throws me of a bit.*
   **Authors:** Thank you for this comment. We agree that the equation defines the wake recovery and not the recovery rate. The text was updated accordingly.

3. **Reviewer:** *At line 232 it is noted that as the power decreases due to the tilt, the wake intensity also decreases which in it of itself already provides higher windspeeds. Would it be possible to discern how much of the power gain for the second turbine comes from the wake redirection and how much from the fact less power is extracted from the incoming wind?*
   **Authors:** Yes, and indeed this point is addressed later in the same section. The vertical speed profiles shown in Fig. 9 allow one to appreciate the vertical displacement caused by tilting, but also show the effects caused by the same tilting on speed deficit. A brief explanation accompanies the figure, providing the necessary interpretation.

4. **Reviewer:** *A (maybe naive) question I have in general is how far (in terms of angle) is the turbine platform combination from tipping over. Could such a scenario exist with a sudden drop in wind speed/thurst force or an onset of large waves, especially if the variation in wind or waves is faster than the pumping system.*
   **Authors:** We added a comment on tip over in Sect. 4. However, a complete analysis on the stability of the system in extreme conditions (including faults) has not been conducted in this work. We have noted this point explicitly both in Sect. 4 and in the conclusions.

5. **Reviewer:** *On page 16, line 279 it says: ...rotor to a slightly faster of slower wind speed. I think of should be or?*
   **Authors:** Thank you, this was indeed a typo, which has now been corrected.

6. **Reviewer:** *On the same page, I get the impression that the labels in figure 13b are mixed up (or the bars). The percentages mention in the text for the cluster power match the bars with the Upstream xlabel and vice versa for cluster.*

**Authors:** Thank you for the comment. Indeed, the labels were mixed up, and they have now been corrected.

**Reviewer #2**

1. **Reviewer:** *I am concerned with the ΔP (tilt angle) curve reported in Figure 8a. As mentioned in the manuscript, previous work on wake steering where the rotor hub is tilted or yawed shows curves of the type P=P0 (cos(angle))p which, as such, are very "flat" near the reference position with no effective yaw or tilt (see e.g. Fig. 10 in the Nanos et al. Torque 2020 paper). This is not what is observed in Fig.8a where the slope of the ΔP (tilt angle) curve is large and strongly discontinuous through the zero-tilt. The effect of the vertical displacement of the hub in a sheared mean wind, mentioned in the discussion of Fig. 8a, does not explain the behavior of the curve as it would be associated to a non-zero but continuous slope of the curve through zero-tilt; indeed, in the chosen configuration, the effect of the vertical displacement of the hub is to induce a negative ΔP for a decreased hub height (negative tilt, negative floater pitch) but a positive ΔP for an increased hub height (positive tilt, positive floater pitch) which is not observed in Fig.8a. Furthermore, with this type of curve, the cosinus-power-law fit is highly questionable (as would be probably apparent by comparing the fit to the actual curve in Fig 8a) ans so is the fitted exponent p=3.5. The authors should clarify this issue which is of primary importance for the subsequent discussion of tilt-induced power change of the cluster.*
   **Authors:** Thank you for pointing this out. We re-ran all simulations (which is one of the reasons why it took so long to prepare the revised version of the paper, in addition to the first author having left the group after defending his Ph.D.) and found that the previous ones had not been run for a long enough time span to reach steady state. In addition, the previous simulations contained an error that caused the nacelle not to be properly oriented when the turbine was tilted. The new simulations indeed show the expected flat behavior of the curve. The plots and text have been updated accordingly. Since the verification of the CFD simulations required a significant effort from another team member (Simone Tamaro), we added this person to the list of authors.

2. **Reviewer:** *The study includes a detailed analysis of the loads experienced by the unwaked tilted turbine (that would be the most upwind one in a wind plant). It would be interesting to know also the loads experienced by the downwind fully waked turbine in the case where it is not tilted, but possibly also in a tilted case that would be expected if additional turbines were added in the column. It would be great if the authors could add this analysis to the revised paper but if they can't, they should at least emphasize in the conclusions that additional work is needed to estimate the loads experienced by the waked floating turbine.*
   **Authors:** If the downstream turbine is not tilted, the loads will be the same that would be experienced by that same turbine installed with a conventional fixed bottom solution. We agree that the study of a tilted and waked turbine would be interesting, but such a study is outside of the scope of this preliminary analysis. As suggested, we modified the conclusions explaining this limitation of the present study.

3. **Reviewer:** *The stability bounds on the tilt angle that can be accessed with the proposed technique should be clearly mentioned/discussed, possibly by showing/discussing at least the hydrostatic restoring moment curve as a function of the tilt (or pitch) angle. Also, the approximate position of the center of gravity should be reported in figures 11 and 12, where the center of flotation is shown.*
   **Authors:** Thank you for the comment. Generally speaking, floating platforms are very stable structures and the pitch angles used here, up to 20 degrees, are in reality rather small. Indeed, the hydroaeroelastic simulations at maximum pitch showed that the platform is stable even in an extreme

sea state, which would never happen in reality because tilt control would not be used in adverse weather conditions. However, it is true that a more extensive study of the system stability would be necessary, including also faults of critical system components. This point was noted and made clearer in the revised version of the manuscript, both in Sect. 4 and in the conclusions.

The approximate position of the center of gravity was added to Fig. 11 but not to Fig. 12, where it would roughly coincide with the center of flotation; this was noted in the text.

We have taken the opportunity to make several small editorial changes to the text, in order to improve readability. A revised version of the manuscript is attached to the present reply, with the main changes highlighted in red (deletions) and blue (additions).

Best regards.
The authors

[revised manuscript text omitted]